# Enhanced 3T3-L1 Differentiation into Adipocytes by Pioglitazone Pharmacological Activation of Peroxisome Proliferator Activated Receptor-Gamma (PPAR-γ)

**DOI:** 10.3390/biology11060806

**Published:** 2022-05-24

**Authors:** Catarina Teixeira, André P. Sousa, Inês Santos, Ana Catarina Rocha, Inês Alencastre, Ana Cláudia Pereira, Daniela Martins-Mendes, Pedro Barata, Pilar Baylina, Rúben Fernandes

**Affiliations:** 1Laboratory of Medical and Industrial Biotechnology (LABMI), Porto Research, Technology, and Innovation Center (PORTIC), Polytechnic Institute of Porto (IPP), 4200-374 Porto, Portugal; catarinateixeira126@gmail.com (C.T.); andre.mp.sousa2@hotmail.com (A.P.S.); nene9717@gmail.com (I.S.); anacmr13@hotmail.com (A.C.R.); ana.bqa@gmail.com (A.C.P.); daniela.m.m.mendes@gmail.com (D.M.-M.); pedro.barata@gmail.com (P.B.); 2Institute of Research, Innovation in Health (i3S), University of Porto, 4200-135 Porto, Portugal; ines.soares77@gmail.com; 3Department of Health Sciences and Functional Biology (FBUVigo), Faculty of Biology, University of Vigo, 36310 Vigo, Spain; 4Unit of Biochemistry (FMUP), Department of Biomedicine, Faculty of Medicine, University of Porto, 4200-319 Porto, Portugal; 5Faculty of Health Sciences (FCS), Clinical Studies Center (CECLIN-HEFP), Fernando Pessoa Hospital, University Fernando Pessoa (UFP), 4420-096 Porto, Portugal; 6School of Health (ESS), Polytechnic Institute of Porto (IPP), 4200-072 Porto, Portugal

**Keywords:** 3T3-L1, pioglitazone, PPAR-γ, adipocyte differentiation

## Abstract

**Simple Summary:**

Pioglitazone is a potent activator of PPAR-γ, a transcriptional factor that is involved in insulin sensibilization and adipocyte differentiation. Here, we propose an optimized methodology for adipocyte differentiation that is critical for secretome release. To achieve this goal, different concentrations of pioglitazone (0–10 µM) were tested and the adipocyte lipidic accumulation was studied. The secretome was then incubated with prostatic cells and macrophages, and the aggressiveness and expression of the inflammatory cytokines were evaluated. We concluded that pioglitazone enhanced adipocyte differentiation and secretome production, making this secretome an excellent adiposity study model.

**Abstract:**

Despite the primary function of pioglitazone in antidiabetic treatment, this drug is a potent inducer of PPAR-γ, a crucial receptor that is involved in adipocyte differentiation. In this work, we propose an optimized methodology to enhance the differentiation of 3T3-L1 fibroblasts into adipocytes. This process is crucial for adipocyte secretome release, which is fundamental for understanding the molecular mechanisms that are involved in obesity for in vitro studies. To achieve this, a pioglitazone dose-response assay was determined over a range varying from 0 to 10 µM. Lipid accumulation was evaluated using Oil-Red-O. The results showed that 10 µM pioglitazone enhanced differentiation and increased secretome production. This secretome was then added into two cell lines: PC3 and RAW264.7. In the PC3 cells, an increase of aggressiveness was observed in terms of viability and proliferation, with the increase of anti-inflammatory cytokines. Conversely, in RAW264.7 cells, a reduction of viability and proliferation was observed, with a decrease in the overexpression of pro-inflammatory cytokines. Overall, the present work constitutes an improved method for adipocyte secretome production that is suitable for experimental biology studies and that could help with our understanding of the molecular mechanisms underlying adiposity influence in other cells.

## 1. Introduction

Pioglitazone is an insulin-sensitizing drug from the thiazolidinediones family [1] that strongly activates the transcription factor peroxisome proliferator-activated receptor gamma (PPAR-γ) [2,3,4,5,6,7]. Due to its biological effects, pioglitazone is widely used as a type 2 diabetes (T2D) treatment, and several studies have shown that PPAR-γ activation can be directly correlated with insulin sensibilization and the regulation of lipidic metabolism (Figure 1) [2,4,6]. Moreover, pioglitazone has also been shown to improve beta-cell function and nonalcoholic steatohepatitis/nonalcoholic fatty liver disease, as well as microalbuminuria reduction [8]. On the other hand, pioglitazone has also been associated with fluid retention, bone fractures, and fat weight gain [8].

PPAR-γ is fundamental in adipogenesis and adipocyte differentiation, accumulating several functions [7,9,10]. This receptor is involved in the process of lipid signaling, which allows lipids to be transported from metabolic organs, including the liver and skeletal muscle, to white adipose tissue (WAT) [7,10]. Moreover, it modulates the expression of adipokines to promote adipocyte differentiation, specifically by increasing the expression of adiponectin and decreasing leptin expression [7,10,11]. Furthermore, this transcription factor is involved in cholesterol and triglyceride plasma regulation, which indirectly relates to cardiovascular comorbidities such as atherogenesis and coronary heart disease [9]. Additionally, PPAR-γ regulates the inflammation observed in WAT, not only by polarizing M1 macrophages, but also by reducing the concentration of these cell types [7]. The PPAR family comprises other important receptors such as PPAR-α, which is likewise involved in adipogenesis, although to a lesser extent, and PPAR-δ, which acts as a constitutive gene [12].

The activation of PPAR-γ by pioglitazone seems to play a role in adipocyte differentiation [13], apoptosis, and hypertrophy, by increasing adiponectin expression [14], as well as the control of inflammatory cascades via the reduction of tumor necrosis factor α (TNF-α) [2,3,7] and interleukin 6 (IL-6) expression [6,7]. Nevertheless, the biological mechanisms of PPAR-γ activation by pioglitazone remains unestablished. 

Herein, we proposed a methodology to evaluate the optimal exposure to pioglitazone, and the levels that better enhance pre-adipocyte differentiation to mature adipocytes. Mature adipocytes release a myriad of molecules with metabolic, immune, and signaling significance, known as the secretome. Improving the adipocyte secretome may be useful for several experimental conditions. Our group has been using the secretome from mature adipocytes as an in vitro model for obesity. In particular, our research with the secretome has been used to understand the mechanisms underlying the effects of obesity in asthma [15] and in cancer cells such as breast and prostate cancer [16,17]. 

This work may be a contribution to biomedical scientists that are dissecting the molecular mechanisms underlying obesity and associated disorders, using an improved pharmacological method for secretome production. 

## 2. Materials and Methods

The experiment was performed following the diagram presented in Figure 2. 

### 2.1. 3T3-L1 Adipocytes Differentiation

A 3T3-L1 mouse pre-adipocytes cell line (American Type Culture Collection (ATCC^®^) CL-173™, Manassas, VA, USA) was maintained in Dulbecco’s modified Eagle’s medium (DMEM, with 4.5 g/L glucose and L-Glutamine, P0103, VWR, Biowest, Nuaillé, France) supplemented with 10% NewBorn Calf Serum (VWR, Biowest, S0400, France), 1% penicillin/streptomycin (P/S) (L0010, VWR, Biowest, France) and 1.5 g/L sodium bicarbonate (S5761, Sigma-Aldrich, Taufkirchen, Germany), at 37 °C in a humidified atmosphere containing 5% CO_2_, and allowed to reach confluence. After confluence (day 0), the medium was replaced by Differentiation Medium I (DMI), which contained 0.25 µM isobutylmethylxanthine (IBMX; I5879, Sigma-Aldrich, Germany), 2 µM insulin (I6634, Sigma-Aldrich, Germany), and 1 µM dexamethasone (D4902, Sigma-Aldrich, Germany) supplemented with 10% fetal bovine serum (FBS; S1520, VWR, Biowest, France) and 1% P/S. After five days (Day 5), the medium was changed to Differentiation Medium II (DMII): DMEM, supplemented only with 10% FBS and 2 µM insulin. On day 10, the culture was washed twice with phosphate-buffered saline (PBS) and a basal medium (DMEM without any serum/supplementation) was added. On day 15, the medium was changed to a new fresh basal medium (BDM). The supernatant (secretome) in contact with the cells from the 16th to the 20th day post-differentiation was harvested, centrifuged at 3000× *g* for 5 min to discard the debris, and stored at −80 °C for subsequent assays. An optimization for adipocyte differentiation and the consequent accumulation of lipid droplets was performed by adding a range of pioglitazone (0, 1.0, 2.5, 5.0, 7.5, and 10.0 µM) to the DMI.

### 2.2. Evaluation of Lipid Droplets Accumulation via the Oil-Red-O Technique

The amount of lipid droplets accumulated at the end of the differentiation process (Day 15) was assessed via the Oil-Red-O technique, according to the Sigma Aldrich protocol. Briefly, after a previous wash with PBS, the cells were fixed with formaldehyde solution (10%) (F8775, Sigma-Aldrich, Germany) and incubated for 30 min. Then, the formaldehyde was discarded, and the cells were washed twice with water and next incubated with 60% isopropanol (I9516, Sigma-Aldrich, Germany) for 5 min. After this, a stock solution of Oil-Red-O (O0625, Sigma-Aldrich, Germany) previously prepared by reconstituting this component in 100% isopropanol was used to prepare a filtered working solution (3 parts of stock solution to 2 parts of water) that was added to the cells for 20 min. Then, hematoxylin (GHS132, Sigma-Aldrich, Germany) was applied for 1 min, after the removal of the working solution. At last, the hematoxylin was removed, and the cells were washed with water. 

### 2.3. Adipocyte RNA Extraction and Gene Expression via RT-qPCR

3T3-L1 adipocytes were cultured in a 6-well plate at a density of 4 × 10^5^ cells/mL, and the differentiation process occurred as described in Section 2.1. The negative control was also performed (undifferentiated cells). Then, the cells were washed with 1× PBS, detached and centrifuged at 950 rpm at 37 °C for 3 min, and the total RNA was extracted using Lab-Aid 824s Nucleic Acid Extraction System (Zeesan, Fujian, China). The concentration and 260/280, and 260/230 ratios of each sample were accessed using the µDrop™ Plate and Thermo Scientific™ Multiskan SkyHigh Microplate Spectrophotometer, Waltham, MA, USA. Gene-specific oligonucleotide primers (Table 1) and probes were designed using PrimerBlast and purchased from Integrated DNA Technologies, Europe. RT-qPCR detection was conducted using the One-step NZY RT-qPCR Probe kit, ROX (NZYTech, Lisboa, Portugal) under the following cycle conditions: incubation at 50 °C for 20 min (reverse transcriptase action), 95 °C for 3 min (polymerase activation) and 40 cycles of 95 °C/10 s and 60 °C/40 s. Relative quantification was determined using the 18S housekeeping gene (reference gene) via the 2^−ΔΔCt^ method. The assay was conducted in duplicates.

### 2.4. Cell Cultures

The PC3 cell line was cultured in RPMI-1640 medium (VWR, Biowest, P0860-N10L, Riverside, MO, USA) and the RAW264.7 cell line (American Type Culture Collection (ATCC^®^) TIB-71, USA) was cultured in DMEM medium. Both cell lines were supplemented with 10% fetal bovine serum (Gibco, Life Technologies, 10270, Waltham, MA, USA) and 1% penicillin/streptomycin (Gibco, Life Technologies, 10270, USA), and maintained at 37 °C in a humidified chamber containing 5% CO_2_. 

### 2.5. Evaluation of the Adipocyte Secretome for the Cellular Viability (MTT) and Proliferation (BrdU) of the PC3 and RAW264.7 Cell Lines 

PC3 and RAW264.7 cells were seeded in a 96-well plate at a density of 1 × 10^5^ cells/well with the respective complete medium. At 70% confluence, the medium was replaced by a medium without FBS, and cells were subjected to starvation for 24 h. After starvation, the cell medium without FBS was supplemented with 10%, 50%, and 100% adipocyte secretome (Section 2.1). Following a 24 h and a 48 h incubation, the cells were washed with 1× PBS and MTT (3-(4,5-dimethylthiazol-2-yl)-2,5-diphenyl tetrazolium bromide) (Thermofisher Scientific, 11319089, Lisbon, Portugal), and BrdU (5-bromo-2′-deoxyuridine) (Abcam, ab126556, Boston, MA, USA) assays were performed. In the MTT assay, cells were incubated with MTT solution at a final concentration of 0.5 mg/mL for 3 h and lysed with DMSO, followed by the measurement of absorbance at 570 nm using a Thermo Scientific™ Multiskan SkyHigh Microplate Spectrophotometer. In the BrdU assay, BrdU was added at the same time as the adipocyte secretome and incubated at 37 °C; after 24 or 48 h, the solution was removed, and the cells were fixed for 30 min. Then, an anti-BrdU antibody marked with an enzyme was added for 90 min at room temperature. At the end of this time, the cells were washed and the substrate was added and incubated for 15 min, and then the absorbance was read at 450 and 690 nm with a Thermo Scientific™ Multiskan SkyHigh Microplate Spectrophotometer. The assays were performed in triplicate, and the mean and the standard deviation values for each experiment were calculated.

### 2.6. Evaluation of the Adipocyte Secretome in the Inflammation of the PC3 and RAW264.7 Cell Lines

Both cell lines were cultured in a 6-well plate at a density of 4 × 10^5^ cells/mL, and after treatment with the adipocyte secretome, the RNA was extracted using the same extraction system and method that was also used in Section 2.3. The inflammatory state was then assessed via the expression of inflammatory genes (IL-1β, IL-4, IL-6, and IL-10) using gene-specific oligonucleotide primers (Table 2) that were designed using PrimerBlast and purchased from Integrated DNA Technologies, Europe. RT-qPCR detection was conducted using the One-Step NZYSpeedy RT-qPCR Green Master Mix (NZYTech), under the following cycle conditions: incubation at 50 °C for 20 min (reverse transcriptase action), 95 °C for 3 min (polymerase activation), and 40 cycles of 95 °C/10 s and 60 °C/40 s. The relative quantification was determined as a -fold increase of the gene of interest compared with the 18S housekeeping gene (reference gene) using the 2^−ΔΔCt^ method. The assay was conducted in duplicate.

### 2.7. Statistical Analysis

The results were expressed as the mean ± standard error of the mean (SEM). Data were analyzed with GraphPad Prism 8.0. The analysis of three or more conditions, and more specifically, the analysis of the MTT, BrdU, and RT-qPCR assays, was conducted using one-way ANOVA with Dunnett’s multiple comparisons test, for which the *p* values of <0.05 were considered to be statistically significant. 

## 3. Results

### 3.1. Pioglitazone Increases Lipid Droplet Accumulation in Differentiated Adipocytes

In order for a better yield for lipid droplet accumulation and for an optimal differentiation of 3T3-L1 pre-adipocytes, an optimization of the differentiation protocol was performed via the addition of 0, 1.0, 2.5, 5.0, 7.5, and 10.0 µM concentrations of pioglitazone to the DMI. The morphological differences between the pre-adipocytes and the differentiated adipocytes are demonstrated in Figure 3A,B. The differentiation level was first evaluated using the amount of lipid droplets accumulated at each of the tested concentrations at the end of the differentiation process. The amount of lipid accumulation increased with the increment of pioglitazone concentration in DMI (Figure 3C–H).

The effectiveness of the differentiation process was assessed using the relative quantification of PPAR-α, δ, and γ gene expression via RT-qPCR. As demonstrated in Figure 4, the expression of PPAR-γ was increased in differentiated adipocytes when compared with PPAR-δ, which are constitutively expressed. 

### 3.2. Pioglitazone Increases PC3 and Decreases RAW264.7 Mitochondrial Metabolic Activity and Cellular Proliferation

The supplementation of the PC3 cell line with the secretome showed an increased tendency, both in viability and proliferation (Figure 5). All of the different percentages of supplementation promoted an increase in the mitochondrial metabolic activity values. Particularly, these values increased from 100 ± 5% (CTRL) to 109 ± 10%, 111 ± 8%, and 107 ± 1% in the 24 h treatments, and from 100 ± 1% (CTRL) to 103 ± 9%, 125 ± 6%, and 124 ± 8% in the 48 h treatments for the supplementation conditions of 10%, 50%, and 100%, respectively. Nevertheless, these differences were not statistically significant. On the other hand, when assessing cellular proliferation, a significant difference occurred under all of the conditions in the 24 h and 48 h treatments, increasing from 100 ± 7% (CTRL) to 185 ± 11%, 190 ± 12%, and 343 ± 38% in the 24 h treatments, and from 100 ± 18% under the CTRL conditions to 285 ± 29%, 292 ± 9%, and 605 ± 85% in the 10%, 50%, and 100% supplementation conditions, respectively. 

In the case of the RAW264.7 cells, supplementation with the adipocyte secretome decreased in a significant manner for both the cellular mitochondrial metabolic activity and proliferation under all of the conditions (Figure 5). More specifically, in cellular viability, in the 24 h treatments, the values decreased from 100 ± 13% under the CTRL condition, to 64 ± 8%, 34 ± 1%, and 32 ± 1%, and from 100 ± 7% (CTRL) to 75 ± 3%, 72 ± 1%, and 71 ± 5% in the 48 h treatments, respectively, under the 10%, 50%, and 100% supplementation conditions. In the proliferation assay case, a decrease occurred in the 24 h treatments, from 100 ± 9% (CTRL) to 84 ± 5%, 57 ± 4%, and 46 ± 2%, and in the 48 h treatments, the values decreased from 100 ± 9% under the CTRL conditions, to 35 ± 2%, 28 ± 1%, and 34 ± 4%, respectively, in the 10%, 50%, and 100% supplementation conditions.

### 3.3. PC3 Revealed a Higher Expression of Anti-Inflammatory than Pro-Inflammatory Cytokine Production, While RAW264.7 Does the Opposite

The negative control (CTRL) in both lines was normalized to 100%, and in comparison with CTRL, the anti-inflammatory cytokine IL-10 presented a higher level of expression than the pro-inflammatory (IL-1β and IL-6) cytokines in the PC3 cell line, with the opposite occurring in the case of the RAW264.6 cell line (Figure 6). More specifically, in the PC3 cells, IL-1 β decreased from 100 ± 7% in the CTRL to 83 ± 3%, 87 ± 1%, and 76 ± 3% (*p* < 0.05) under the 10%, 50%, and 100% supplementation conditions, respectively. The same conditions presented an increase from 100 ± 3% to 114 ± 6% in the 10% supplementation condition, and a decrease to 87 ± 0% and 92 ± 8% in the 50% and 100% supplementation conditions, respectively, for IL-6 expression. In the case of IL-10, this cytokine presented an increase from 100 ± 3% in the CTRL to 140 ± 7% (*p* < 0.001), 147 ± 5% (*p* < 0.0001), and 135 ± 72% (*p* < 0.001) in the 10%, 50%, and 100% supplementation conditions, respectively. 

In the case of RAW264.7, the IL-1β cytokine increased its expression from 100 ± 4% in CTRL to 153 ± 34% (*p* < 0.05), 124 ± 3%, and to 130 ± 15%, and IL-6 gene expression increased from 100 ± 2% under the CTRL condition to 190 ± 3% (*p* < 0.001), 154 ± 2% (*p* < 0.05), and to 163 ± 4% (*p* < 0.05) in 10%, 50%, and 100% supplementation conditions, respectively. IL-10 was the less highly expressed cytokine, presenting in these conditions a decrease from 100 ± 7% in CTRL to 80 ± 2%, 71 ± 1%, and 72 ± 1%. 

## 4. Discussion

Pioglitazone is a powerful activator of PPAR-γ, with an important role in insulin-sensitizing and lipid metabolism. PPAR-γ is involved in adipogenesis and adipocyte differentiation, as well as lipidic signaling, by controlling the transport of lipids from metabolic organs to WAT. This lipidic signaling process in turn involves the expression of adipokines that allow for adipocyte differentiation and the regulation of cholesterol and triglyceride plasma concentration, in addition to the inflammatory state observed in WAT. This study aimed to optimize the adipocytes’ differentiation process by adding pioglitazone to the differentiation medium, and to assess the potential of a secretome (containing molecules secreted by fully differentiated adipocytes to the cell culture) as a simulator of an obese environment. The effectiveness of the differentiation process was assessed through the expression of PPAR genes, and the ability to mimic obesity conditions in the secretome was evaluated through cytometry assays. To understand how different concentrations of pioglitazone modulate adipocyte differentiation, a pre-adipocyte cell line differentiation protocol was evaluated with and without pioglitazone. Through the Oil-Red-O technique, it was verified that the increase of pioglitazone concentration promoted a progressive increase in the accumulation of lipid droplets (Figure 3). According to earlier studies, this output puts in evidence that pioglitazone enhances adipocyte differentiation, hence increasing the differentiation effectiveness [13]. To reinforce this fact, an analysis of gene expression of the three main PPAR genes involved in the differentiation process was employed. The results showed that the expression of PPAR-γ was increased in differentiated cells when compared with the undifferentiated cells. Moreover, when compared with the constitutive gene, PPAR-δ, the expression of PPAR-γ was also significantly elevated. Furthermore, PPAR-α showed a decrease in expression. These findings support the literature evidence pointing to an increase in PPAR-γ expression through its activation by pioglitazone, emphasizing this receptor’s correlation with obesity-related adipogenesis [7].

After verifying the enhancer effect of pioglitazone as an in vitro modulator of adipogenesis (Figure 7), we aimed to evaluate the impact of adipocyte-secreted substances on the culture medium (secretome) in other cell lines’ behavior and the way in which these factors mimic the obese environment. In this manner, two cell lines that are greatly influenced by the obese condition were tested: the first one was a prostatic carcinoma cell line, PC3, which presents a metabolism based on lipidic synthesis and degradation; and RAW264.7, a macrophage cell line that allows for the assessment of the immunomodulation that occurs in the obese environment. Regarding the PC3 cell line, when compared to the CTRL conditions, the results showed an increase in cellular viability and proliferation, with the latter being the condition with more prominent differences for supplementation with the 100% secretome under both the 24 h and 48 h treatments (Figure 5). This result was expected, considering that obesity is a predisposing factor to the development of prostate cancer [18,19]. Interestingly, in RAW264.7, the opposite was observed: both cellular viability and proliferation decreased significantly when compared with CTRL (Figure 5), which could be explained by the polarization of macrophages into the M1 stage under an obesity state [20], where cells lose their replication capabilities, and so, their proliferation. Consequently, the macrophages’ viability decreases, since their lifespan is limited [21].

Considering that both cell lines are directly influenced by the inflammatory state that is observed in obesity [22,23], the assessment of the expression of pro-inflammatory cytokines, namely, IL-1β and IL-6, as well as anti-inflammatory cytokines, such as IL-10, was critical. According to previous findings, the gene expression of these cytokines had opposing effects in PC3 and RAW264.7. Particularly, the anti-inflammatory cytokine IL-10 presented significatively increased gene expression in the PC3 cell line, whereas in RAW264.7 cells, this cytokine was less highly expressed. In the case of the pro-inflammatory cytokines, the results showed an increased degree of expression in RAW264.7, while in PC3, the cells displayed a decrease in expression. Although IL-1β had a higher rate of expression in the macrophage cell line, this increase was only significant for the 10% supplementation condition. In the same cells, the increased expression of IL-6 was significant in all of the tested conditions. In the PC3 cell line, the reduction of both these cytokines’ expression only presented statistical significance for the 100% supplementation condition in the IL-1β gene. These results corroborate the literature findings: in the PC3 cell line, in order to survive and proliferate in a hostile environment, such as the oxidative stress state observed in obesity, the prostate cancer cell line enhances the activation of anti-inflammatory cytokines and reduces the expression of pro-inflammatory cytokines to contradict the inflammatory environment of obesity [24]. In the RAW264.7 cell line, the opposite was observed once the pro-inflammatory cytokines increased their expression in response to the elevated oxidative stress characteristics of obesity; this fact is consistent with the previously proposed polarization of RAW264.7 macrophages into the M1 state, in which pro-inflammatory cytokines are overexpressed [23,25]. 

In conclusion, the effectiveness of adipocyte differentiation that is achieved with this protocol allowed for the production of a secretome that is capable of inducing the same cellular response that was previously described in the literature, even in human cell lines. The huge advantage of this secretome is the relief regarding the ethical issues, as well as regarding the cost of experiments that typically employ human serums. Thus, enhancing adipocyte differentiation by the activation of PPAR-γ with pioglitazone may be a promising methodology for achieving an in vitro model of obesity mimicry. 

The tumoral and inflammatory properties demonstrated by the secretome reflect the obesity-related complications that are seen in the clinical field, proving this to be a good model for obesity in vitro studies. Thus, the novelty of this study was the enhancement of adipocyte differentiation with pioglitazone, which allowed for the approximation of in vitro and in vivo models. 

Nevertheless, further studies are required to deepen the knowledge of secretome composition. Specifically, proteomic and lipidomic analysis, as well as the evaluation of extracellular vesicles and microRNA composition, is crucial. 

## Figures and Tables

**Figure 1 biology-11-00806-f001:**
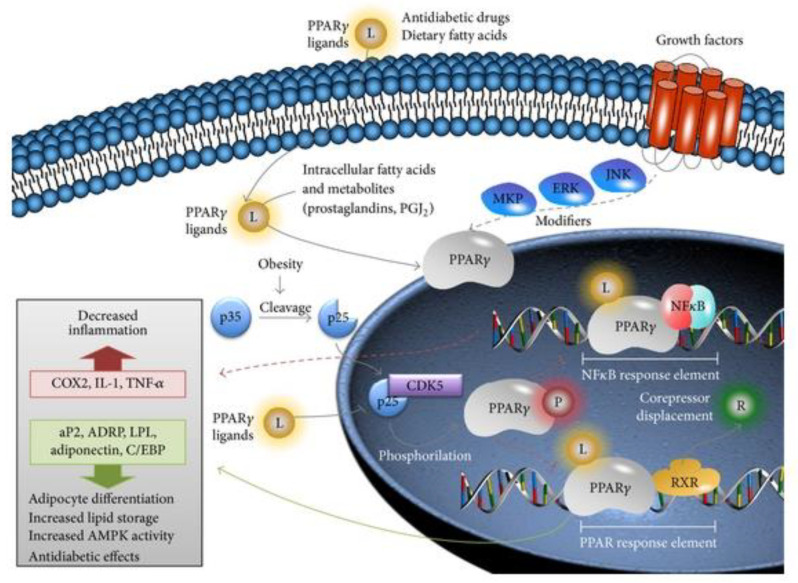
PPAR-γ transduction pathway. PPAR-γ has several extracellular and intracellular ligands that include dietary and bioactive lipids. Given its antidiabetogenic role, some PPAR-γ ligands include antidiabetic drugs such as thiazolidinediones. PPAR-γ is also modulated by several growth factor transduction pathways such as Jnk/Erk/MKP. It is known that the cyclin-dependent kinase 5 (Cdk5) bond to p25 (a product of the cleavage of p35 in an obesity environment) inhibits the PPAR pathway via its phosphorylation. As a transcriptional factor, PPAR binds to RXR (retinol X receptor) to transcribe several genes related to adipocyte differentiation and lipid storage in adipose tissue and that increase insulin sensitivity in peripheral tissues by an indirect increase of AMP kinase activity, as well as several other antidiabetogenic effects. Moreover, PPAR blocks NFB signaling, thus reducing proinflammatory cytokines and inflammation. (https://doi.org/10.1155/2013/401274 (accessed on 15 January 2022)—A. C. Pereira, R. Oliveira, A. C. Castro, and R. Fernandes).

**Figure 2 biology-11-00806-f002:**
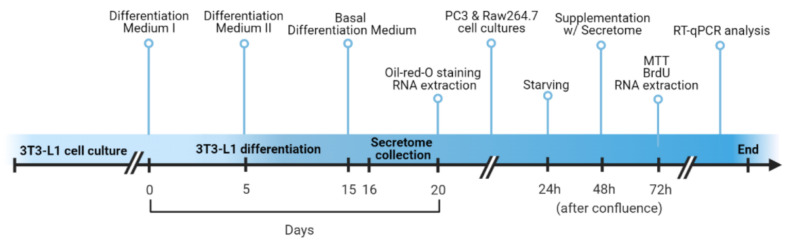
Schematic timeline of the experiment. We started with 3T3-L1 pre-adipocyte differentiation and at the end of the process (between 16 and 20 days), we collected the cell culture medium (secretome) and then we performed Oil-Red-O staining and RNA extraction. At 24 h after confluence, PC3, and Raw264.7 cell lines were submitted to starvation, and 24 h later, they were supplemented with the secretome previously produced by differentiated adipocytes. After that, the viability (MTT), proliferation (BrdU), and RNA extraction and analysis were assessed.

**Figure 3 biology-11-00806-f003:**
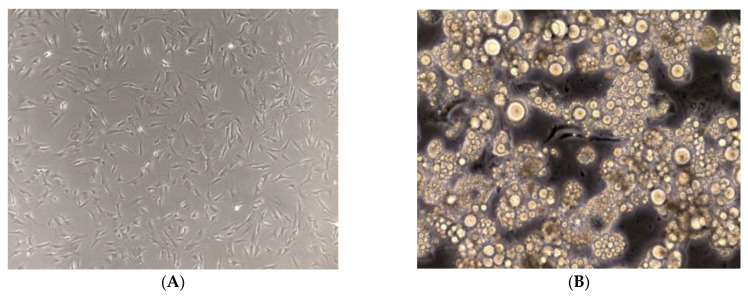
3T3-L1 cell differentiation into mature adipocytes. (**A**) Undifferentiated pre-adipocytes (20× objective); (**B**) Differentiated adipocytes (20× objective). (**C**–**H**) Accumulation of different lipid droplets according to the different pioglitazone levels: 0, 1, 2.5, 5, 7.5, and 10 µM, respectively (10× objective).

**Figure 4 biology-11-00806-f004:**
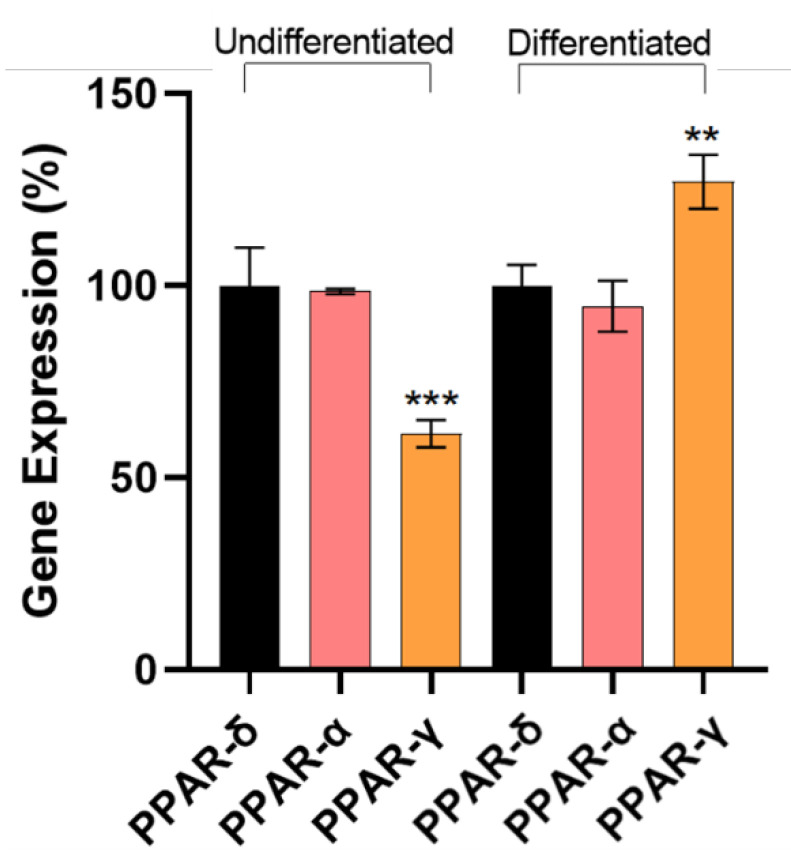
Gene expression of PPAR-α, δ, and γ genes in undifferentiated and differentiated adipocytes. The results were normalized to 100% of PPAR-δ expression. The assay was performed in triplicate. *p* values of < 0.05 were considered statistically significant. ** means *p* < 0.01 and *** means *p* < 0.001.

**Figure 5 biology-11-00806-f005:**
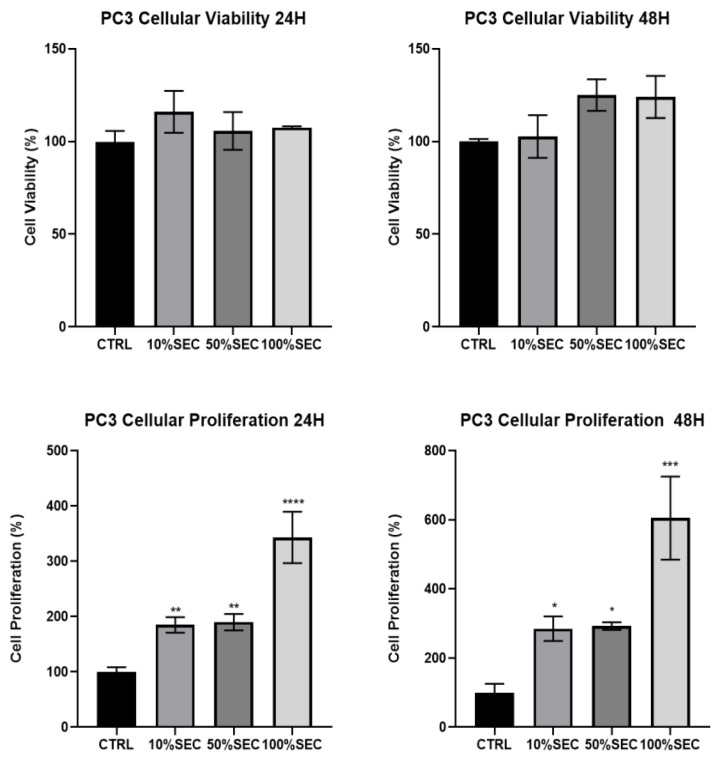
Cellular viability and proliferation of the PC3 and Raw264.7 cell lines after the 24 h and 48 h treatments. The results were compared with the CTRL condition (cells cultured in a culture medium without any serum) that was normalized to 100%. *p* values of <0.05 were considered to be statistically significant: * means *p* < 0.05; ** means *p* < 0.01; *** means *p* < 0.001 and **** means *p* < 0.0001.

**Figure 6 biology-11-00806-f006:**
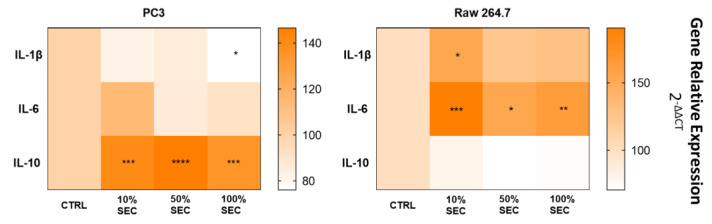
Gene expression of pro-inflammatory (IL-1β and IL-6) and anti-inflammatory (IL-10) cytokines in the PC3 and RAW264.7 cell lines supplemented with 10%, 50%, and 100% of the adipocyte secretome. The results were compared with the CTRL condition (cells cultured with medium without any serum) that was normalized to 100%. *p* values of <0.05 were considered to be statistically significant: * means *p* < 0.05; ** means *p* < 0.01; *** means *p* < 0.001 and **** means *p* < 0.0001.

**Figure 7 biology-11-00806-f007:**
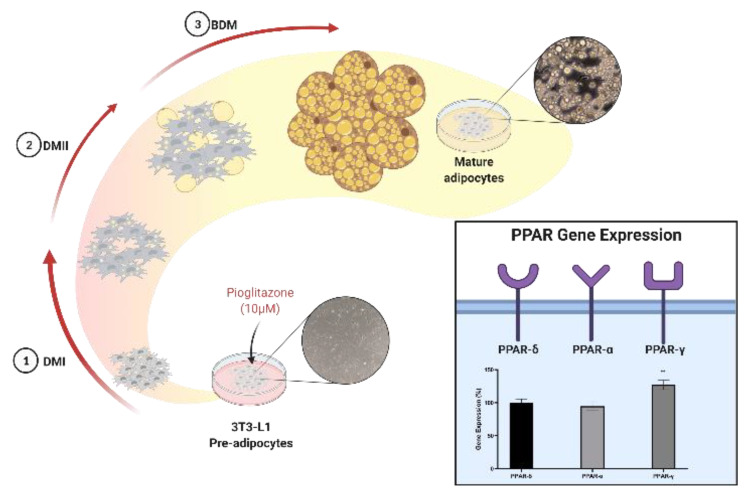
3T3-L1 differentiation into mature adipocytes through pioglitazone action. This drug modulates significatively (** means *p* < 0.01) PPAR family activity to enhance the in vitro differentiation of pre-adipocytes at a concentration of 10 µM.

**Table 1 biology-11-00806-t001:** Gene-specific oligonucleotide primers and probes of PPAR family for RT-qPCR.

Gene	Primer Forward	Primer Reverse	Probe
18S	ACCGCAGCTAGGAATAATGGA	GCCTCAGTTCCGAAAACCA	CY5- ACCGCGGTTCTATTT -BHQ3
PPARα	GCCGAAAGAAGCCCTTACAG	GCCTCAGGGTACCACTAC	JOE-ACATGCGTGAACTCCGT-BHQ1
PPARδ	AGATCCGATCGCACTTCTCA	AGGCGGCAGCCTCAACAT	ROX-AAGGGCTTCTTCCGCC-BHQ2
PPARγ	CCTGCATCTCCACCTTATTATTCTG	CCTTGCATCCTTCACAAGC	FAM-CCTCATGAAGAACCTTCTAACTCCCTCATGGC-BHQ1

**PPARα/δ/γ**: peroxisome proliferator-activated receptor alpha/delta/gamma; **CY5**: Cyanine 5 fluorophore; **JOE**: 5′-Dichloro-dimethoxy-fluorescein fluorophore; **ROX**: carboxy-X-rhodamine; **FAM**: Carboxyfluorescein; **BHQ1/2/3**: Black Hole Quencher ½/3.

**Table 2 biology-11-00806-t002:** Gene-specific oligonucleotide primers of inflammatory cytokines for RT-qPCR.

Gene	Primer Forward	Primer Reverse
18S	ACCGCAGCTAGGAATAATGGA	GCCTCAGTTCCGAAAACCA
IL-1β	ACCTAGCTGTCAACGTGTGG	TCAAAGCAATGTGCTGGTGC
IL-4	GCAGCTGATCCGATTCCTGA	TCCAACGTACTCTGGTTGGC
IL-6	TGTGTGAAAGCAGCAAAGAGG	TTTTCACCAGGCAAGTCTCC
IL-10	TGAAAACAAGAGCAAGGCCG	ATAGAGTCGCCACCCTGATG

**IL-1β/4/6/10**: Interleukine-1β/4/6/10.

## Data Availability

Not applicable.

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
