# Peer review of "Enhanced 3T3-L1 Differentiation into Adipocytes by Pioglitazone Pharmacological Activation of Peroxisome Proliferator Activated Receptor-Gamma (PPAR-γ)"

_biology, 2022, doi:10.3390/biology11060806_

Round 1

Reviewer 1 Report

  1. Introduction

Overall, very informative and clear introduction that gives clarity to the project. Minor spelling and grammatical error that needs polishing in-terms of writing. One of such in line 90, “being” to “been”.

  1. Material and Methods

I think this section requires numbering of the sub-categories, which is missing. However, for this review, I will be number accordingly with order:

Section 2.1:

Good explanation of the design of the differentiation method. Would be clearer to add a schematic timeline of this experiment.

Section 2.2:

Given the notoriety of Oil Red O (ORO) staining with adipocytes, ORO has been now utilized in numerically using metric from spectrophotometry application. Just wanted to know the reasoning that the author did not use this approach?

Section 2.3:

Several elements are missing in this section such as the manner cDNA was synthesized as well as thermal cycles used for qPCR. Furthermore, though the authors mentioned that the assay’s were done in duplicates, how does this reflect on the statistical significance? Were there biological replicates (minimum triplicates) that these experiments were conducted with? I ask because the qPCR results also show n=1, which is not preferable for conducting experiments.

Section 2.4:

Both cell lines use the same media components and maintenance conditions. There is redundancy in this section.

Section 2.5:

Good explanation. However, I would still link a question to section 2.2, given here spectrophotometry was used extensively.

Section 2.6:

I am a bit confused as to the numbering of the sections. However, if the authors mention section 1.3, then I would assume its section 2.3. Similarly, several elements are missing like section 2.3.

Section 2.7:

Were these conducted in three biological replicates (triplicates)? Though ANOVA was performed, there is no explanation as to if P-Values were used.

  1. Results

Section 3.1:

Though there is a clear morphological difference, this does not explain the effects of pioglitazone effect on the yield of lipid accumulation in a dose dependent manner. This would only be better represented if the overall lipid particles were quantified alongside the total cell number. Furthermore, the images need a metric bar.

In regard to the PPAR genes, I could understand why this has been used but this requires more explanation in this section. Furthermore, though the genes were quantified to the house keeping gene 18S, however the manner as how was this quantified in not clear. Another aspect from the figure, is that it indicates n=1, which surprising given that there are error bars. More explanation needed in this section.

Section 3.2 and 3.3:

Overall, very interesting findings, which might be the main landmark of this work. In fact, the main interesting point illustrated in this work is how adipokines by-products from mature adipocytes and therefore secretome can have variable effects on different cell types (the forementioned PC3 and RAW264.7 cell lines). The only thing that needs to be added in these sections are better explanation how the data was generated. The authors mentioned that viability and proliferation were normalized to 100%; however, how the raw data were normalized to this end product is missing. Furthermore, again how many replicates were used in this experiment and what are the values of statistical significance (P-Values) that were observed is missing.

  1. Discussion

In the discussion section, there are several incoherence’s in the write up that needs to be re-written or polished in terms of grammar and spelling mistakes. I highly advise that this needs to be revised and written well.

Content wise, the authors highlighted the significance of pioglitazone effect on adipogenesis in a dose dependent manner utilizing the ORO staining and gene expression; however, the conclusion brought up is yet incomplete. In addition to the need of illustrating ORO quantity numerically, the authors just demonstrated the significance using PPAR isoforms but not other pioneering factors such as CEBP family (specifically alpha and beta; as demonstrated in previous studies PMID: 33345692 and PMID: 32877276). Given the significant machineries associated with factor and adipogenesis, it would be eminent to also get insight into the CEBP family role and the effect of pioglitazone. In addition, the authors highlight the expression of PPAR at the gene levels but the protein level, which would be necessary to get a complementary view in regard to the effect of pioglitazone.

The strength of this section is linked to the hallmark finding in having a conversed effect between two different cell lines (PC3 and RAW264.7) when exposed to the secretome product material of mature adipocytes. Furthermore, linking the pro and anti-inflammatory markers is would great interest into the field of adipocyte differentiation with potential clinical application, which have been evidently highlighted in this study. Yet, again it would be of great interest to link to see if the CEBPs also are involved in this aspect.

Author Response

 Introduction

Overall, very informative and clear introduction that gives clarity to the project. Minor spelling and grammatical error that needs polishing in-terms of writing. One of such in line 90, “being” to “been”.

ANSWER #1

Firstly, I would like to thank you so much for your positive appreciation and comments. All the text was submitted to spelling and grammatical corrections.

Material and Methods

I think this section requires numbering of the sub-categories, which is missing. However, for this review, I will be number accordingly with order:

Section 2.1:

Good explanation of the design of the differentiation method. Would be clearer to add a schematic timeline of this experiment.

ANSWER #2

Thank you for your kind suggestion. An explaining timeline was added in material and methods section.

Section 2.2:

Given the notoriety of Oil Red O (ORO) staining with adipocytes, ORO has been now utilized in numerically using metric from spectrophotometry application. Just wanted to know the reasoning that the author did not use this approach?

ANSWER #3

Thank you so much for your comment. Indeed, we know that that approach is commonly used, however, we didn’t use the spectroscopy because we only wanted to visually demonstrate the phenotypic lipidic accumulation in response to the different concentrations of pioglitazone.

Section 2.3:

Several elements are missing in this section such as the manner cDNA was synthesized as well as thermal cycles used for qPCR. Furthermore, though the authors mentioned that the assay’s were done in duplicates, how does this reflect on the statistical significance? Were there biological replicates (minimum triplicates) that these experiments were conducted with? I ask because the qPCR results also show n=1, which is not preferable for conducting experiments.

ANSWER #4

I'd want to express my gratitude for your constructive feedback. In fact, the absence of thermal cycles may lead the reader confused. We don’t have done a previous step to synthesize cDNA: we used a one-step RT-qPCR and added a first step in the thermal cycle to synthesize the cDNA. As you suggest, we add the thermal cycles to give more clarity to this section of methodology (lines 160-163 and 203-207). Regarding the biological replicates, we conducted the assays (adipocytes differentiation and PC3 and Raw264.7 supplementation) with triplicates, and the evaluation of gene expression by RT-qPCR was done in duplicates for each previous triplicate. The only reason for the gene expression been conducted in duplicates is for monetary reasons. However, we completely understand that the optimal analysis should include at least triplicates.  

Section 2.4:

Both cell lines use the same media components and maintenance conditions. There is redundancy in this section.

ANSWER #5

Thank you so much for your comment. However in this section is explained that, although the supplementation with serum and antibiotics were equal to both cell lines, they were cultured in different culture mediums: PC3 cell line used RPMI-1640 medium, while Raw264.7 was cultured in DMEM. Even so, the text were restructured to make it less repetitive.  

Section 2.5:

Good explanation. However, I would still link a question to section 2.2, given here spectrophotometry was used extensively.

ANSWER #6

Thank you so much for your kind appreciation. As I explained before, we only wanted to see the differences in the phenotype of such cells. In later manuscripts we will take your suggestion into account.

Section 2.6:

I am a bit confused as to the numbering of the sections. However, if the authors mention section 1.3, then I would assume its section 2.3. Similarly, several elements are missing like section 2.3.

ANSWER #7

Thank you so much for your warning. Indeed, it was a typo, already corrected.

Section 2.7:

Were these conducted in three biological replicates (triplicates)? Though ANOVA was performed, there is no explanation as to if P-Values were used.

ANSWER #8

Thank you for your comment. The p-values were added in the statistical analysis section.

Results

Section 3.1:

Though there is a clear morphological difference, this does not explain the effects of pioglitazone effect on the yield of lipid accumulation in a dose dependent manner. This would only be better represented if the overall lipid particles were quantified alongside the total cell number. Furthermore, the images need a metric bar.

In regard to the PPAR genes, I could understand why this has been used but this requires more explanation in this section. Furthermore, though the genes were quantified to the house keeping gene 18S, however the manner as how was this quantified in not clear. Another aspect from the figure, is that it indicates n=1, which surprising given that there are error bars. More explanation needed in this section.

ANSWER #9

Thank you very much for your observations. As I explained before, we only used the oil-red-o staining to see phenotypical accumulation of lipid particles. After that, we choose the condition that seems to present a higher quantity of lipidic accumulation, to assess the PPAR genes expression.

As explained in 2.3 section, the quantification of PPAR genes expression regarding the constitutive expression of 18s housekeeping gene was done by 2-ΔΔCt method, as explained in lines 162 and 206. In this method is implicit that the 18S is used as reference gene for the relative expression quantification.

Section 3.2 and 3.3:

Overall, very interesting findings, which might be the main landmark of this work. In fact, the main interesting point illustrated in this work is how adipokines by-products from mature adipocytes and therefore secretome can have variable effects on different cell types (the forementioned PC3 and RAW264.7 cell lines). The only thing that needs to be added in these sections are better explanation how the data was generated. The authors mentioned that viability and proliferation were normalized to 100%; however, how the raw data were normalized to this end product is missing. Furthermore, again how many replicates were used in this experiment and what are the values of statistical significance (P-Values) that were observed is missing.

ANSWER #10

Thank you so much for your kind appreciation. As referred in 2.5 section, the assay was conducted in triplicates. The mean value of the negative control (CTRL) (a condition that was not supplemented with secretome) triplicates was standardized to 100%, functioning as cellular basal growth. The other conditions (supplementation with 10%, 50% and 100% of secretome) were normalized to the CRTL. The statistical significance were referred in statistical analysis as well as in the results section.  

Discussion

In the discussion section, there are several incoherence’s in the write up that needs to be re-written or polished in terms of grammar and spelling mistakes. I highly advise that this needs to be revised and written well.

ANSWER #11

Thank you so much for your advertisement. Grammatical and spelling corrections were done in all manuscript.

Content wise, the authors highlighted the significance of pioglitazone effect on adipogenesis in a dose dependent manner utilizing the ORO staining and gene expression; however, the conclusion brought up is yet incomplete. In addition to the need of illustrating ORO quantity numerically, the authors just demonstrated the significance using PPAR isoforms but not other pioneering factors such as CEBP family (specifically alpha and beta; as demonstrated in previous studies PMID: 33345692 and PMID: 32877276). Given the significant machineries associated with factor and adipogenesis, it would be eminent to also get insight into the CEBP family role and the effect of pioglitazone. In addition, the authors highlight the expression of PPAR at the gene levels but the protein level, which would be necessary to get a complementary view in regard to the effect of pioglitazone.

ANSWER #12

I'm thankful for your suggestions. The analysis of other factors like CEBP family is of extremely importance. However, in this preliminary study, we only wanted to see the direct activation of PPAR genes family by pioglitazone. Also, the expression analysis of the correspondent proteins involved in the adipogenic process will be considered in further studies. Aside from that, we don't have the primers needed to test these genes, and the process of obtaining them would take longer than the time we have to complete the major modifications.

The strength of this section is linked to the hallmark finding in having a conversed effect between two different cell lines (PC3 and RAW264.7) when exposed to the secretome product material of mature adipocytes. Furthermore, linking the pro and anti-inflammatory markers is would great interest into the field of adipocyte differentiation with potential clinical application, which have been evidently highlighted in this study. Yet, again it would be of great interest to link to see if the CEBPs also are involved in this aspect.

ANSWER #13

We also think that it would be of great interest explore CEBPs role in a future research project.

Reviewer 2 Report

Pioglitazone is a potent activator of the transcription factor PPAR-γ; accordingly, it enhances insulin sensitivity. Therefore, Pioglitazone is used as an effective therapeutic agent for type 2 diabetes.

This study aims to establish Pioglitazone as an adipogenic activator and determine how Pioglitazone treated adipocytes secretome affects two cell lines, PC3 and RAW264.7.

However, the study is poorly designed, and the manuscript has severe language issues. I recommend rewriting the manuscript.

Some of the main concerns are mentioned here. 

  1. In the introduction, lines 50-82 are unnecessary descriptions of PPAR-γ. Authors should write on Pioglitazone instead.
  2. To establish adipogenesis, authors should consider at least 5-6 adipogenic markers besides PPAR-γ, e.g., CEBP-β, adiponectin, FABP4.
  3. The authors' main objective was on the adipocyte secretome that might cause cellular proliferation and viability; however, they have not done any experiment to show what types of secretome might cause these changes.
  4. It appears to me that the control image (Figure 2B) of differentiated adipocytes accumulated more lipid than any Pioglitazone treated adipocytes (figures 2C-2H), which is inconsistent with figure 6.
  5. Figure 3 needs control with every PPAR-γ, PPAR-α & PPAR-δ expression.
  6. Figure 4 needs positive and negative control with every graph.
  7. The authors should explain the novelty of this research. 

Author Response

REVIEWER #2:

Pioglitazone is a potent activator of the transcription factor PPAR-γ; accordingly, it enhances insulin sensitivity. Therefore, Pioglitazone is used as an effective therapeutic agent for type 2 diabetes.

This study aims to establish Pioglitazone as an adipogenic activator and determine how Pioglitazone treated adipocytes secretome affects two cell lines, PC3 and RAW264.7.

However, the study is poorly designed, and the manuscript has severe language issues. I recommend rewriting the manuscript.

ANSWER #1

Thank you for your constructive comments. As requested by reviewer #1, the entire document was evaluated and modified in terms of spelling and grammar.

Some of the main concerns are mentioned here. 

  1. In the introduction, lines 50-82 are unnecessary descriptions of PPAR-γ. Authors should write on Pioglitazone instead.
  2. To establish adipogenesis, authors should consider at least 5-6 adipogenic markers besides PPAR-γ, e.g., CEBP-β, adiponectin, FABP4.
  3. The authors' main objective was on the adipocyte secretome that might cause cellular proliferation and viability; however, they have not done any experiment to show what types of secretome might cause these changes.
  4. It appears to me that the control image (Figure 2B) of differentiated adipocytes accumulated more lipid than any Pioglitazone treated adipocytes (figures 2C-2H), which is inconsistent with figure 6.
  5. Figure 3 needs control with every PPAR-γ, PPAR-α & PPAR-δ expression.
  6. Figure 4 needs positive and negative control with every graph.
  7. The authors should explain the novelty of this research. 

ANSWER #2

Thank you for all your helpful suggestions.

  1. Your suggestion was very important and an explanation on Pioglitazone benefits and risks was included in lines 50-53.
  2. As explained to reviewer #1, the analysis of other factors like CEBP family is of extremely importance. However, in this preliminary study, we only wanted to see the direct activation of PPAR genes family by pioglitazone. Also, we don't have the primers needed to test these genes, and the process of obtaining them would take longer than the time we have to complete the major modifications.
  3. This was an exploratory study that aimed to understand the effect of pioglitazone as an enhancer of adipocytes differentiation and the consequent effect of the released adipokines in some cell lines. In future studies, it will certainly address important questions such as an in-depth analysis of secretome in terms of proteomics, lipidomics, extracellular vesicles and micro RNA’s. Even so, a paragraph elucidating further studies was added in lines 371-374.
  4. The figures 2A and 2B intends to demonstrate the difference between undifferentiated and differentiated adipocytes, respectively, before being stained with oil red o. The figures 2C-2H demonstrates the dose-response effect of pioglitazone in lipidic accumulation of fully differentiated adipocytes (after staining with oil red o).
  5. We add the PPAR’s genes expression of the negative control condition (undifferentiated cells) to the graph of figure 4.
  6. The negative control is already included (CTRL).
  7. The novelty of the study was added in lines 368-370.

Reviewer 3 Report

In this paper authors present an interesting work where they proposes the use of the cellular secretome to in vitro mimic the inflammatory state characteristic of obesity. For this propose, 3T3-L1 cells are differentiated into mature adipocytes after treatment with the PPARγ-Agonist Pioglitazone. The work is interesting, although some aspects should be better explained and further investigated.

1. Line 210-215 (Figure3): The effectiveness of differentiation process was assessed by a relative quantification of PPAR-α, δ and γ genes expression by RT-qPCR, using the PPAR-δ as control. However, it is not clear that the expression of this gene is constitutive in 3T3-L1, as this receptor in involved in different cellular metabolic functions (for a review see doi: 10.3390/ijms19113339). Thus, qPCR should demonstrate the overexpression of PPARg after treatment with different doses of Pioglitazone, in comparison with untreated cells. Other PPARg-target genes involved in lipid accumulation, as Fasn, Adipoq or Plin1 should be also quantified in order to demonstrate the effect over adipogenesis.

2. Treatment of RAW264 cells with the secretome seems to be cytotoxic. Moreover, it has not well explained why the cytokine secretion is opposite to PC3 cells.

3. Int his line, polarization of RAW264.7 macrophages into M1 state after secretome exposure has not completely demonstrated by the authors, it is only one hypothesis. Authors should demonstrate this polarization with, at least, gene expression analyses.

4) Authors have not explained whether a murine (3T3-L1) secretome could affect in different way depending on the human/murine of the recipient cell.

5) Figure 4: Figures 4A and 4B could be merged into a single figure using a linear graph, expressing the cell viability (y-axis) and dose of treatment (x-axis) in two lines (24 and 28h). The same can be applied for figure 4C and D and figure 4E and F.

Other minor points:

Line 87: A reference is missed.

Figures 2, 4, 5 and 6 are not referenced along the text.

Author Response

REVIEWER #3:

In this paper authors present an interesting work where they proposes the use of the cellular secretome to in vitro mimic the inflammatory state characteristic of obesity. For this propose, 3T3-L1 cells are differentiated into mature adipocytes after treatment with the PPARγ-Agonist Pioglitazone. The work is interesting, although some aspects should be better explained and further investigated.

  1. Line 210-215 (Figure3): The effectiveness of differentiation process was assessed by a relative quantification of PPAR-α, δ and γ genes expression by RT-qPCR, using the PPAR-δ as control. However, it is not clear that the expression of this gene is constitutive in 3T3-L1, as this receptor in involved in different cellular metabolic functions (for a review see doi: 10.3390/ijms19113339). Thus, qPCR should demonstrate the overexpression of PPARg after treatment with different doses of Pioglitazone, in comparison with untreated cells. Other PPARg-target genes involved in lipid accumulation, as Fasn, Adipoq or Plin1 should be also quantified in order to demonstrate the effect over adipogenesis.
  2. Treatment of RAW264 cells with the secretome seems to be cytotoxic. Moreover, it has not well explained why the cytokine secretion is opposite to PC3 cells.
  3. In this line, polarization of RAW264.7 macrophages into M1 state after secretome exposure has not completely demonstrated by the authors, it is only one hypothesis. Authors should demonstrate this polarization with, at least, gene expression analyses.
  4. Authors have not explained whether a murine (3T3-L1) secretome could affect in different way depending on the human/murine of the recipient cell.
  5. Figure 4: Figures 4A and 4B could be merged into a single figure using a linear graph, expressing the cell viability (y-axis) and dose of treatment (x-axis) in two lines (24 and 28h). The same can be applied for figure 4C and D and figure 4E and F.

ANSWER #1

Firstly, I would like to thank so much your kind and positive appreciation.

  1. There are many articles, including the one that is referenced with number 12 in the manuscript (doi: 10.1210/me.2004-0539) that demonstrate that PPAR- δ is a good control of differentiation process because despite being involved in the adipogenesis process, its expression does not vary during this process. Even so, the expression of PPAR genes in undifferentiated cells was added in the graph of figure 4. As I explained to the other revisors,
  2. In lines we put the hypothesis that the reduction of macrophages viability and proliferation may be explained by the loss of the replication capability when they after their polarization, and so present a limited lifespan.
  3. Your suggestion is very crucial, however, in this preliminary study we only evaluate the expression of pro-inflammatory cytokines.
  4. There are already many articles (ex.: https://doi.org/10.1074/mcp.m600217-mcp200) that demonstrate that 3T3-L1 can be used as model of obesity in human cells.
  5. Despite we are thankful for your suggestion, in our point of view, the way that we present the results are more simple, visual, and intuitive.

Other minor points:

  1. Line 87: A reference is missed.
  2. Figures 2, 4, 5 and 6 are not referenced along the text.

ANSWER #2

Thank you so much for your advertisements.

  1. This mistake was corrected.
  2. Highlights to the mentioned figures were added in the text, namely in the discussion section.

Round 2

Reviewer 1 Report

Overall a good and interesting study. Looking forward to linking it to future work.

Reviewer 2 Report

  1. Regarding the author's comment 1: It is recommended to rewrite the introduction again. Avoid figure 1 and focus on Pioglitazone in the introduction. 
  2. Regarding comment 2: If authors want to establish adipogenesis, better take your time for these experiments. 
  3. Regarding comment 3: Figure 2 (now figure 3), it is not scientific to put figure 3B with other oil red O staining images (3C-3H) and conclude anything from there. It is highly recommended to have done oil red O staining for all conditions.  
  4. Maybe the authors could not catch my comment 4; please read carefully. It appears that the control image (Figure 3B) of differentiated adipocytes accumulated more lipid than any Pioglitazone treated adipocytes (figures 3C-3H), which is inconsistent with figure 7". 
  5. It is assumed from figure 3 that there is no effect of Pioglitazone on differentiated adipocytes.
  6. Figure 4 only shows adipogenic gene expressions between undifferentiated vs. differentiated adipocytes. What would happen if Pioglitazone exposes to differentiated adipocytes. In fact, Pioglitazone was the author's primary concern.
  7. No way can authors validate figure 7 from all experiments.

Author Response

Response in attachment.

Kind regards.

Reviewer 3 Report

I recommend this version for publication.

Author Response

Response in attachment.

Kind regards.

This manuscript is a resubmission of an earlier submission. The following is a list of the peer review reports and author responses from that submission.